# Mid-Infrared Spectroscopy as a New Tool for Ruling Out Spontaneous Bacterial Peritonitis: A Proof-of-Concept Study

**DOI:** 10.3390/biomedicines11030838

**Published:** 2023-03-09

**Authors:** Marwin A. Farrugia, Maëna Le Corvec, Christophe Renou, Jean-Baptiste Nousbaum, Dann J. Ouizeman, Olivier Sire, Olivier Loréal, Hugues Tariel, Jérôme Bernard, Thierry Piche, Albert Tran, Hafid Ait-Oufella, Luce Landraud, Philippe Gual, Rodolphe Anty

**Affiliations:** 1Service d’Hépatologie, Centre Hospitalier Universitaire, Hôpital Archet 2, 151 Route de Saint-Antoine, 06000 Nice, France; 2DIAFIR, Avenue Chardonnet, Parc Lorans 26J, 35000 Rennes, France; 3Centre Hospitalier de Hyères, Bd Maréchal Juin, 83400 Hyères, France; 4Centre Hospitalier Régional Universitaire de Brest, 2, Avenue Foch, 29609 Brest, France; 5IRDL, UMR CNRS 6027, Université Bretagne Sud, 56000 Vannes, France; 6NUMECAN, UMR INSERM 1241, Centre Hospitalier Universitaire, 35000 Rennes, France; 7INSERM, U1065, C3M, Université Côte d’Azur, 06000 Nice, France; 8Service de Réanimation Médicale, AP-HP, Hôpital Saint-Antoine, INSERM UMR 991, Inserm U970, Paris Research Cardiovascular Center, Université Pierre et Marie Curie, 75012 Paris, France; 9Université Paris Cité and Université Sorbonne Paris Nord, INSERM, IAME, Louis Mourier Hospital, 92700 Paris, France

**Keywords:** spontaneous bacterial peritonitis, ascites fluid, infection, cirrhosis, mid-infrared spectroscopy

## Abstract

Background and aims: A highly sensitive and specific point-of-care method for diagnosing spontaneous bacterial peritonitis (SBP) is currently lacking. The objective of the present study is to evaluate the diagnostic value of a rapid, easy-to-use, mid-infrared fiber evanescent wave spectroscopy (MIR-FEWS) method for ruling out SBP. Patients and Methods: Cirrhotic patients (*n* = 256) at five centers in France were included for suspected SBP or for the scheduled evacuation of ascites fluid. The mid-infrared spectrum of 7 µL of an ascites fluid sample was recorded using a MIR-FEWS system. To define a model for the diagnosis of SBP, the patients were divided into a calibration group (*n* = 170) and a validation group (*n* = 86). Results: Most of the patients were male (71%). The mean age was 60.25 years. Alcohol-related liver disease was the most common cause of cirrhosis. SBP was observed in 18% of the patients. For the diagnosis of SBP in the calibration and validation groups, respectively, the model gave areas under the receiver operating characteristic curves of 0.87 and 0.89, sensitivities of 90% and 87%, specificities of 78% and 80%, positive predictive values of 48% and 50%, negative predictive values of 97% and 96%, positive likelihood ratio of 4.09 and 4.35, negative likelihood ratio of 0.13 and 0.16, Youden index of 0.68 and 0.67, and correct classification rates of 80% and 81%. Conclusion: The results of this proof-of-concept study show that MIR-FEWS is a highly sensitive diagnostic method for ruling out SBP. The method warrants further investigation.

## 1. Introduction

Spontaneous bacterial peritonitis (SBP) is a frequently lethal complication in patients with ascites and cirrhosis. Although the mortality rate has decreased over the last 30 years, it remains between 23% and 58% [1,2]. As with other infections in cirrhotic patients, earlier treatment is associated with a lower likelihood of death [3].

Several point-of-care (POC) tests (such as urine dipsticks in ascites fluid) have been developed but are not sensitive enough [4]. It has been suggested that sophisticated ascites cytokine profiles can help the clinician to manage patients with suspected SBP [5,6,7]. However, given the lack of international guidelines on the use of these tools, there is an unmet need for a specific, sensitive POC test for SBP [1].

Mid-infrared (MIR) spectroscopy measures the vibrational interactions between a sample and mid-infrared light. The MIR spectrum of a patient’s biofluid may constitute a metabolic fingerprint that can be used for diagnostic purposes [8]. Several mid-infrared spectroscopy techniques have been developed. In mid-infrared fiber evanescent wave spectroscopy (MIR-FEWS), the sample to be analyzed is deposited on an optical fiber capable of transmitting MIR light. In the present study, a sensor with a chalcogenide glass optical fiber and an appropriate spectrometer (DIAFIR, Rennes, France) were used [8]. The MIR spectra were analyzed with specific validated software, in order to give the clinician a result within 15 min; no engineering know-how is required. This MIR-FEWS technique has recently been described as a POC diagnostic for infections in patients with septic arthritis [9,10].

The objective of the present study is to develop a new spectral model for ruling out SBP in a French multicenter study of a cohort of patients with ascites and cirrhosis.

## 2. Materials and Methods

### 2.1. Patients

An initial group of 123 patients with ascites and cirrhosis was recruited prospectively at four centers (Brest, Hyères, Nice, and Monaco) between 2010 and 2014, during a French multicenter study funded by the French national hospital-based clinical research program (forming the PHRC cohort; ClinicalTrials.gov NCT01193426) [11]. The goal was to define a specific cytokine profile produced during SBP [11]. Clinical and laboratory data were collected for all patients. Laboratory data included analyses of blood and ascites samples.

A second group of 133 patients with ascites and cirrhosis admitted to an intensive care unit (at Saint-Antoine University Medical Center, Paris, France; the Paris cohort) was studied prospectively. The study was approved by an institutional review board (CPP IV Ile de France, Paris, France; reference 2014/04NI) and performed in accordance with the principles of the Declaration of Helsinki. The goal was to define specific ascitic cytokines during the development of SBP, and the corresponding results have been published elsewhere [11]. Detailed clinical and laboratory data were not available for the Paris cohort. 

The main inclusion criteria applied to the two cohorts were as follows: Aged 18 or over, social security coverage, the provision of informed consent, and admission for the treatment of ascites or complications of cirrhosis. The exclusion criteria applied to the two cohorts were abdominal surgery within the previous month, the presence of chylous ascites or ascites not related to portal hypertension (pancreatic ascites, hemoperitoneum, ascites observed during acute heart failure, peritoneal tuberculosis, and hepatocellular carcinoma), and severe obesity (body mass index ≥35 kg/m^2^).

Spontaneous bacterial peritonitis was diagnosed as a polymorphonuclear (PMN) leukocyte count in the ascites fluid ≥250/mm^3^, in line with the current guidelines, [12,13].

Ascites cultures were obtained for all patients. 

All patients gave their written, informed consent to participate.

### 2.2. Ascites Sample Analysis

For each patient in the PHRC and Paris cohorts, ascites samples were collected on the day of hospital admission, in dry sterile tubes (BD Vacutainer^®^ EST^TM^ N°362725 13 × 75 mm). The cell count was determined using a magnification microscope with the KOVA™ Glasstic™ Slide 10 with Grids device. The samples were collected from the two cohorts between 2010 and 2014. After centrifugation at 5000 g/min, the supernatants were frozen, stored at −80 °C at each investigating center, and subsequently (in 2015) analyzed centrally using MIR-FEWS.

### 2.3. Mid-Infrared Fiber Evanescent Wave Spectroscopy

#### 2.3.1. Acquisition and Pre-Treatment of Spectra

The MIR absorbance spectra (frequency range: 4000–800 cm^−1^) were recorded for 15 min, using LS23 single-use sensors and a SPID^TM^ FT-IR spectrometer (DIAFIR, Rennes, France). A FEWS infrared sensor was placed in the spectrometer, the background signal was recorded, and 7 μL of ascites fluid were deposited on the sensor for acquisition of the spectrum.

Only spectra that passed quality controls (sufficient signal amplitude at background and acquisition stage, final signal-to-noise ratio, and the water to other elements signal ratio) were selected. In order to reduce physical and environmental sources of bias, the spectra were preprocessed and normalized.

The data’s homogeneity (i.e., the possible presence of outliers) was checked by visual inspection of a principal component analysis plot.

#### 2.3.2. MIR-FEWS Analysis

The objective of the statistical analyses of the spectra was to identify patients with an ascites fluid PMN leukocyte count <250/mm^3^ (i.e., no SBP) or ≥250/mm^3^ (i.e., SBP). The patients’ spectra were randomized 2:1 into a calibration group and a validation group. The two randomized groups had the same proportions of patients with SBP.

Firstly, the most informative spectral variables (i.e., those that best distinguish between a PMN leukocyte count <250 vs. ≥250) were selected. The initial set of 615 spectral variables was reduced by applying a least absolute shrinkage and selection operator, a random forest algorithm, and factor-adjusted discriminant analysis. Secondly, a logistic regression model (hereafter referred to as the spectral model) was defined. It produced a score ranging from zero (fluid PMN leukocyte count <250/mm^3^) to one (fluid PMN leukocyte count ≥250), which corresponded to the probability of identifying a patient with an ascites fluid PMN leukocyte count ≥250/mm^3^. The optimal threshold was chosen according to the “closest.topleft” method [14] i.e., the point closest to the top-left part of the area under the curve plot with perfect sensitivity or specificity.

The model’s diagnostic performance with a given threshold was characterized by calculating the area under the receiver operating characteristic curve (AUROC), the sensitivity, the specificity, the negative predictive values (NPV), the positive predictive value (PPV), positive likelihood ratio (LR+), negative likelihood ratio (LR−), Youden index and the correct classification rate.

### 2.4. Statistical Analysis

Continuous variables were expressed as the median [interquartile range]. The Shapiro–Wilk test was used to determine whether or not data were normally distributed. Groups were compared using Student’s *t*-test (for normally distributed data) or the Mann–Whitney U-test. Categorical data were assessed using Fisher’s exact test. Spearman’s test was used to study the correlations between variables.

The 95% confidence intervals for the performance indicators and the AUROCs were estimated from 2000 bootstrap replicates (using the pROC package in R software [14]). The threshold for statistical significance was set to *p* < 0.05. All statistical analyses were performed with R software [15].

## 3. Results

### 3.1. Characteristics of the Study Participants

Spontaneous bacterial peritonitis was found in 47 (18%) of the 256 patients. The characteristics of the study participants with and without SBP are summarized in Table 1. Data on most of the clinical and laboratory variables were only available for the PHRC cohort (*n* = 123).

The majority of the patients were male (71%), and the study population’s mean age was 60.3 years. The causes of cirrhosis were variously alcohol (*n* = 83, 67.4%), viral hepatitis and alcohol (*n* = 19; 15.4%), viral hepatitis (*n* = 8; 6.5%), non-alcoholic fatty liver disease (*n* = 4, 3.3%), biliary cirrhosis (*n* = 3, 2.4%), and other causes (*n* = 6; 5%). There were no significant differences between the calibration and validation groups (Appendix A).

### 3.2. The Spectral Model’s Ability to Identify SBP

All the spectra passed the quality controls, and the principal component analysis did not reveal any outliers. The ascites fluid-specific spectral model was defined with four spectral variables (the absorption values at four different wavelengths). The model’s AUROCs for the diagnosis of SBP were 0.87 and 0.89 in the calibration and validation groups, respectively (Figure 1).

At the optimal threshold, the calibration model had a sensitivity of 90%, a specificity of 78%, a PPV of 48%, an NPV of 97%, a Youden index of 0.68, and a correct classification rate of 80%. The validation model gave a sensitivity of 87%, a specificity of 80%, a PPV of 50%, an NPV of 96%, a Youden index of 0.67, and a correct classification rate of 81% (Table 2). A comparison of the AUROCs and the scores did not detect any significant differences between the PHRC cohort and the Paris cohort (Figure 2).

The spectral model’s scores were significantly correlated with the PMN leukocyte count in the calibration samples (*r* = 0.44; *p* < 0.001), the validation samples (*r* = 0.64; *p* < 0.001), and the samples as a whole (*r* = 0.48; *p* < 0.001).

None of the studied variables accounted for the misidentification of some patients, i.e., the five false-negative patients (three in the calibration group and two in the validation group; Appendix A) and the 44 false-positive patients (30 in the calibration group and 14 in the validation group). A positive culture did not appear to have a significant influence on the spectral model’s score (Appendix A). An analysis of cases as a function of the bacteria involved would not have been statistically robust, given the small size of each subgroup.

## 4. Discussion

In the present study, an innovative MIR-FEWS technique was used to rule out SBP in cirrhotic patients. The AUROCs in the calibration and validation groups were 0.87 and 0.89, respectively.

Spontaneous bacterial peritonitis is a frequent and highly lethal complication of end-stage liver disease [12]. According to the guidelines, specific treatment of SBP must be initiated as soon as possible [12,13]. This is only possible if the diagnosis is actively sought by obtaining an ascites fluid sample and an immediate cytology assessment. The result should then be sent as soon as possible to the clinician, so that he/she can initiate appropriate treatment. Problems at any of these steps can delay the initiation of specific treatment for SBP. In this situation, a highly sensitive and specific POC test would be helpful for the immediate initiation of treatment of SBP. Several POC technologies (including urine test strips) have been assessed. However, a large prospective study showed that the Multistix test strip was not sensitive or specific enough [16]. Use of a second-generation test strip improved the specificity but not the sensitivity [17]. Specific markers in the ascites or serum have also been investigated; these include various cytokines, calprotectin, lactoferrin, and triggering receptor expressed on myeloid cells 1 (TREM-1) [11,18,19,20,21]. A synthesis of the results of these studies and the current study is presented in Appendix A. Up to now, a specific, easy-to-measure marker is not yet available, and the international guidelines do not recommend any diagnostic markers [12].

The SPID^TM^ MIR-FEWS system can be easily operated by the clinician. Its value in the diagnosis of septic arthritis has already been demonstrated in a pilot study and a large prospective study [9,10]. It has been suggested that MIR-FEWS will be of clinical value in other fields of hepatology, such as the diagnosis of non-alcoholic steatohepatitis or the prognostic evaluation of patients with ascites and cirrhosis [8,22].

Our present results suggest that MIR-FEWS is useful and has a very high NPV for ruling out SBP within 15 min. A negative result can thus avoid the initiation of inappropriate antibiotic treatment. Of course, a cytology assessment of the ascites fluid sample should be performed as soon as possible. The diagnostic accuracy appeared to be equivalent or superior to that of other tests described in the literature in this clinical setting (Appendix A). The negative predictive value was high, which enabled us to unambiguously detect patients without SBP.

The present study has several strengths. Firstly, the patients were prospectively included, and the ascites fluid samples were prospectively collected. Secondly, the patients came from several general hospitals and two university hospitals. Thirdly, the large number of patients made it possible to establish validation and calibration groups and thus avoid overfitting. 

The study also had some limitations. Firstly, the MIR-FEWS technique was not tested as a true POC test (i.e., at the time when the patients were being cared for in the various investigating centers; the analyzed samples had been collected and frozen several years previously). Secondly, the PPV was low; half of the patients were wrongly considered to have SBP. Thirdly, long-term data on the morbidity and mortality of the study population were lacking. It would be valuable to investigate the possible impact of this rapid diagnostic approach on the three-month morbidity and mortality of patients in a prospective multicenter study.

The present proof-of-concept study highlighted the diagnostic value of an innovative MIR-FEWS technique. To validate this new technology as an accurate POC test, our results will now have to be confirmed in a large, prospective, multicenter study. The test’s specificity could perhaps be increased by combining the spectral model with one or several clinical or laboratory variables.

## 5. Conclusions

An innovative MIR-FEWS technique might be of value for rapidly ruling out SBP (within 15 min) and with a high degree of sensitivity. A large, prospective, multicenter study is now required to confirm these findings. MIR-FEWS might be a new tool to help the clinician manage patients with ascites, cirrhosis and SBP; as emphasized in the treatment guidelines, every hour gained is precious.

## Figures and Tables

**Figure 1 biomedicines-11-00838-f001:**
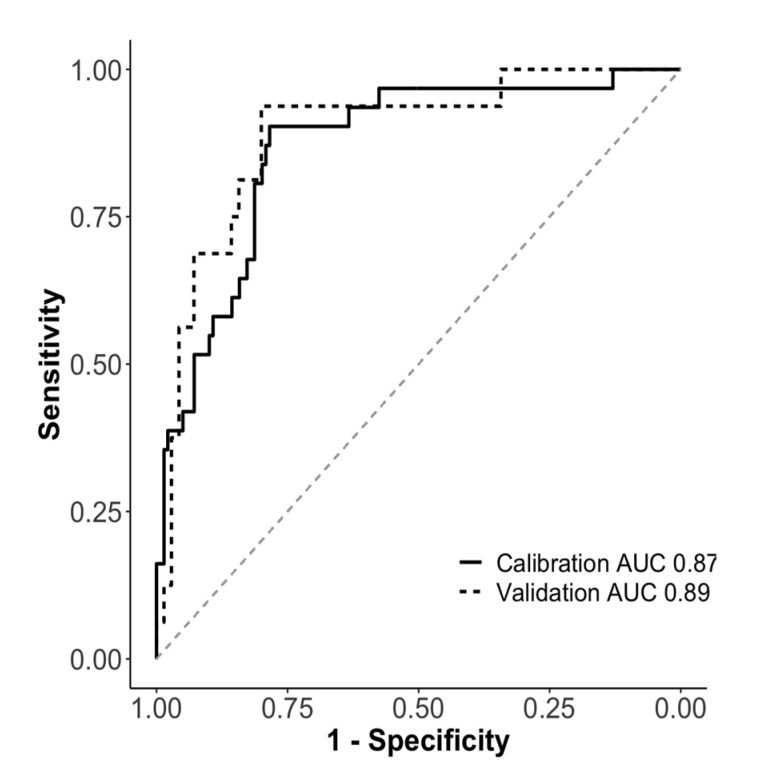
The spectral model’s AUROC for the calibration group (solid line) and the validation group (dotted line).

**Figure 2 biomedicines-11-00838-f002:**
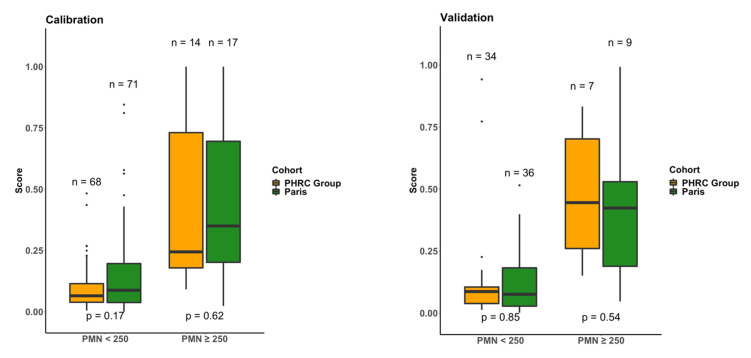
Comparisons of the spectral model’s scores for the calibration and validation groups in the PHRC and Paris cohorts, according to the presence or absence of SBP (ascites fluid PMN leukocyte count <250/mm^3^ vs. ≥250/mm^3^).

**Table 1 biomedicines-11-00838-t001:** Characteristics of the study participants, as a function of the ascites fluid PMN leukocyte count.

Variables	PMN Leukocyte Count <250/mm^3^ (*n* = 209)	PMN Leukocyte Count ≥250/mm^3^ (*n* = 47)	*p*
Ascites score ^#^	3 [2–3]	3 [2–3]	0.498
Encephalopathy score ^#^	1 [1–1]	1 [1–1]	0.661
Prothrombin time (%) ^#^	58.5 [50–73.8]	46 [27–61]	0.013
Albumin (g/L) ^#^	29 [26–32]	27 [25–29]	0.049
Bilirubin (µmol/L) ^#^	41 [19–85.5]	55 [37–170]	0.090
Blood platelet count (G/L) ^#^	127 [81–192]	82 [62–142]	0.037
Blood creatinine (µmol/L) ^#^	69 [54–107]	89 [64–134]	0.189
Child–Pugh score^#^	9 [8–11]	11 [10–13]	0.002
Age (years) ^#^	60 [51.2–69.8]	62.5 [50.8–65.2]	0.999
% males ^#^	68	86	0.119
PMN leukocyte count (/mm^3^)	6 [1–26]	2256 [793.5–4776]	<0.001
Positive ascites fluid culture (%)	4	47	<0.001
Distribution according to recruitment centers (Brest/Hyères/Monaco/Nice/Paris) (%)	5/10/1/32/52	6/6/2/30/56	0.882

^#^ data from the PHRC cohort only (*n* = 123). Quantitative data are expressed as the median [interquartile range]. PMN: Polymorphonuclear.

**Table 2 biomedicines-11-00838-t002:** The model’s ability to identify patients with an ascites fluid PMN leukocyte count ≥250/mm^3^.

	*n*	PMN *≥* 250/mm^3^ (n)	AUROC	Threshold	Se	Sp	PPV	NPV	CC Rate	Youden Index	LR+	LR−
**Calibration group**	**170**	**31**	**0.87 [0.80–0.94]**	**0.1629**	**90** **[77–100]**	**78** **[71–85]**	**48** **[40–57]**	97 **[94–100]**	**80** **[75–86]**	**0.68** **[0.48–0.85]**	**4.09 [3.03–6.06]**	**0.13 [0.01–0.28]**
Paris cohort	88	17	0.84 [0.73–0.95]		94 [82–100]	72 [61–82]	44 [36–55]	98 [94–100]	76 [67–84]	0.66 [0.43–0.82]	3.36 [2.27–5.07]	0.08 [0–0.29]
PHRC cohort	82	14	0.91 [0.83–0.98]		86 [64–100]	85 [76–94]	54 [41–74]	97 [92–100]	85 [77–93]	0.71 [0.4–0.94]	5.73 [3.26–12.12]	0.16 [0–0.43]
**Validation group**	**86**	**16**	**0.89 [0.80–0.98]**		**87** **[69–100]**	**80** **[70–88]**	**50** **[39–65]**	**96** **[92–100]**	**81** **[73–89]**	**0.67** **[0.39–0.88]**	**4.35 [2.75–7.97]**	**0.16 [0–0.40]**
Paris cohort	45	9	0.86		89 [67–100]	72 [58–86]	44 [32–64]	96 [89–100]	75 [62–87]	0.61 [0.25–0.86]	3.18 [1.79–6.14]	0.15 [0–0.52]
PHRC cohort	41	7	0.93 [0.85–1]		86 [57–100]	88 [76–97]	60 [42–87]	97 [91–100]	88 [78–97]	0.74 [0.33–0.97]	7.17 [3.37–NA]	0.16 [0–0.51]

The 95% CIs are given in square brackets. PMN: Polymorphonuclear leukocyte count; Se: sensitivity; Sp: Specificity; PPV: PPositive predictive value; NPV: Negative predictive value; CC: Correct classification; LR+: Positive likelihood ratio; LR−: negative likelihood ratio; AUROC: Area under the receiver operating characteristic curve; NA: Not applicable.

## Data Availability

The data presented in this study are available in this article and the Appendix A.

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
