# Peer review of "Mid-Infrared Spectroscopy as a New Tool for Ruling Out Spontaneous Bacterial Peritonitis: A Proof-of-Concept Study"

_biomedicines, 2023, doi:10.3390/biomedicines11030838_

Round 1
Reviewer 1 Report
The manuscript by Marwin A. Farrugia et al. describe results from two ancillary cohorts to evaluate mid-infrared spectroscopy.
The manuscript is of interest but deserve to be extensively revised.
Global : italicize "ie". Introduce all acronyms before their first use. Numbers less than 12 must be spelled out. The whole manuscript must be written in the passive tense.
Title : The concept of "proof of concept study" must be highlighted.
MEthods : the slection criteria are crucial and cannot only be described by referring to the main manuscript. In front of this particular inclusion process, al rests must be stratifiedaccording to the particularities of both the cohorts.
As required by the French regulation, Has the CPP been consulted for this third cohort (at least substancial modification to the first two cohorts)?
Methods: how were the ascites collected (collection medium? delivery time?) The authors should describe the homogeneity/heterogeneity of the biological techniques used.
Method: What were the positive/negative controls?
Method: How was the multiplication of statistical tests considered (Bonferroni's approaches?)?
Method : Paragraph 2.4 : The sentence line 143 -145 should be deleted (not appropriate in an article). The sentence line 141-142 should be moved to the end of the manuscript in the appropriate part.
Results: "4 spectral variables" means ? give details.
Results: Prefer truthfulness ratios to prevalence-dependent predictive values. especially for comparison. Replace "well-classified rates" by Youden index.
Results : it would be interesting to get the results according to the bacteria involved (last paragraph of part 3).
Author Response
The manuscript by Marwin A. Farrugia et al. describe results from two ancillary cohorts to evaluate mid-infrared spectroscopy.
We thank you for your guidance in improving this work.
The manuscript is of interest but deserve to be extensively revised.
The manuscript has been fully reviewed by a professional English language editor.
Global : italicize "ie". Introduce all acronyms before their first use. Numbers less than 12 must be spelled out. The whole manuscript must be written in the passive tense.
We have made the modifications as suggested. The manuscript has been fully reviewed by a professional English language editor.
Title : The concept of "proof of concept study" must be highlighted.
The title has been changed like this: « Mid-infrared spectroscopy as a new tool for ruling out spontaneous bacterial peritonitis: a proof-of-concept study ».
Methods : the selection criteria are crucial and cannot only be described by referring to the main manuscript. In front of this particular inclusion process, al rests must be stratifiedaccording to the particularities of both the cohorts.
We have added the inclusion and exclusion criteria in the method section (lines 93 to 99).
As required by the French regulation, Has the CPP been consulted for this third cohort (at least substancial modification to the first two cohorts)?
The CPPs were consulted during the constitution of the 2 cohorts (PHRC and Paris). The current study was constructed from the samples of the two previous cohorts. Patients and CPPs were informed that the samples would be kept for future studies on the diagnosis of spontaneous bacterial peritonitis (NCT01193426: "Diagnosis of spontaneous bacterial peritonitis" and PPC IV Ile de France, Paris, France; reference 2014/04NI).
This was specified in the "Patients" section.
Methods: how were the ascites collected (collection medium? delivery time?) The authors should describe the homogeneity/heterogeneity of the biological techniques used.
We added this paragraph in “Ascites sample analysis”: “For each patient in the PHRC and Paris cohorts, ascites samples were collected on the day of hospital admission. The cell count was determined automatically. The two cohorts’ samples were collected between 2010 and 2014. After centrifugation at à 5000g/min, the supernatants were frozen, stored at -80°C at each participating site, and subsequently (in 2015) analyzed centrally using MIR-FEWS.” (Lines 107 to 111).
Method: What were the positive/negative controls?
There was no positive/negative control. The endpoint was the accepted threshold of PNN ≥ 250/mm3. Patients ≥250 PNN/mm3 were compared with those with PNN <250 /mm3.
Method: How was the multiplication of statistical tests considered (Bonferroni's approaches?)?
Our approach was to try several learning machine techniques on our spectra database, and to keep the most efficient one. Each technique, whereas it is FADA, LASSO or RF is designed to reduce the number of parameters considering possible correlations.
Spectral data being correlated, Bonferroni test is not appropriate is this situation. Validation is done by applying the model, built using a batch of 2/3 of the patients, called calibration group, on the last third of the cohort, the validation group. Patients are set apart in order to have similar distribution of patient characteristics in both calibration and validation groups.
Then, selection between the three method is based on their relative performance for the validation cohort.
Method : Paragraph 2.4 : The sentence line 143 -145 should be deleted (not appropriate in an article). The sentence line 141-142 should be moved to the end of the manuscript in the appropriate part.
We have made the modifications as suggested.
Results: "4 spectral variables" means ? give details.
The 4 spectral variables are Mid InfraRed signal absorption at 4 distinct wavelengths. The said wavelength not being to specific elements as MIR spectroscopy does identify molecular liaisons and not molecules. However, these wavelength cover vibrations of liaison existing in glucose and its metabolites. So, we can assume that the MIR spectra is correlated to bacteria metabolic activity within ascites liquid.
We have added this explanation in the manuscript (line 185).
Results: Prefer truthfulness ratios to prevalence-dependent predictive values. especially for comparison. Replace "well-classified rates" by Youden index.
We have listened to your advice and have modified Table 2 (page 6).
Results : it would be interesting to get the results according to the bacteria involved (last paragraph of part 3).
These results would have little statistical significance because of the heterogeneity of the bacteria identified and the small number of patients who had the same bacterial species. We specified this at the end of this paragraph.
We hope you will enjoy this new version of our manuscript.
We thank you again for your guidance in improving our article.
Yours faithfully.
Marwin FARRUGIA
Reviewer 2 Report
New diagnostic method, for a common complication that occurs in ascitic patients. Few comments:
Line 143-144: Due to the lack of availability of one of the group leaders (self-promotion, management of the Covid-19 health crisis, and then personal reasons), new analyses and then the writing of the article were only recently performed.........I am not sure what is the value of these details!!
The sample size is needed in the statistical analysis section.
Line 7-8: 5 false negative patients (3 in the calibration group and 2 in the validation group; Table A2) and 44 false positive patients (30 in the calibration group and 14 in the validation group)......Do you think this important data has to be added to the drawbacks of the study???
Line 52-53: We cannot determine the potential impact on patient survival between those managed conventionally and those with diagnostic testing using MIR-FEWS......Clinically, just after taking the ascitic fluid sample, a double-dose antibiotic started immediately without waiting for the ascitic fluid sample analysis results. Of course, if there is a way for a fast-specific test, this will be wonderful. So, I am not sure that there will be any mortality difference between those managed conventionally and those with diagnostic testing using MIR-FEWS.
Author Response
New diagnostic method, for a common complication that occurs in ascitic patients. Few comments:
We thank you for your guidance in improving this work.
Line 143-144: Due to the lack of availability of one of the group leaders (self-promotion, management of the Covid-19 health crisis, and then personal reasons), new analyses and then the writing of the article were only recently performed.........I am not sure what is the value of these details!!
We have made the modifications as suggested.
The sample size is needed in the statistical analysis section.
We did not establish a calculation of the number of patients to be included, because this study was a proof-of-concept study.
Confirmation of the study requires a large prospective multicenter study with a calculation of the number of patients to be included in order to determine the best diagnostic performance and to improve the positive predictive value.
The value of conducting a new prospective multicenter study was highlighted in the discussion.
Line 7-8: 5 false negative patients (3 in the calibration group and 2 in the validation group; Table A2) and 44 false positive patients (30 in the calibration group and 14 in the validation group)......Do you think this important data has to be added to the drawbacks of the study???
Thank you for this comment. We have indeed addressed in the discussion the low positive predictive value in the weaknesses of our study (lines 53-54, page 8).
Line 52-53: We cannot determine the potential impact on patient survival between those managed conventionally and those with diagnostic testing using MIR-FEWS......Clinically, just after taking the ascitic fluid sample, a double-dose antibiotic started immediately without waiting for the ascitic fluid sample analysis results. Of course, if there is a way for a fast-specific test, this will be wonderful. So, I am not sure that there will be any mortality difference between those managed conventionally and those with diagnostic testing using MIR-FEWS.
We share your opinion in cases where the clinical suspicion of ascites infection is strong. However, the diagnosis is sometimes uncertain in clinical routine (patient with few symptoms). If the MIR-FEWS technique tested in a large prospective cohort showed a good NPV and PPV, one could hope for a faster initiation of antibiotic therapy in patients with no or few symptoms. This could translate into reduced morbidity or mortality in the short term. This is speculation and hypothesis currently. We propose to retain this idea in the paragraph on limitations of the current study.
We hope you will enjoy this new version of our manuscript.
We thank you again for your guidance in improving our article.
Yours faithfully.
Marwin FARRUGIA
Round 2
Reviewer 1 Report
The manuscript has been well-improved regarding to my preivous comments. Nevertheless, some remains to be adressed.
Methods: how were the ascites collected (collection medium? delivery time?) The authors should describe the homogeneity/heterogeneity of the biological techniques used.
We added this paragraph in “Ascites sample analysis”: “For each patient in the PHRC and Paris cohorts, ascites samples were collected on the day of hospital admission. The cell count was determined automatically. The two cohorts’ samples were collected between 2010 and 2014. After centrifugation at à 5000g/min, the supernatants were frozen, stored at -80°C at each participating site, and subsequently (in 2015) analyzed centrally using MIR-FEWS.” (Lines 107 to 111).
--> Give details about the automates, the sample media.
Method: What were the positive/negative controls?
There was no positive/negative control. The endpoint was the accepted threshold of PNN ≥ 250/mm3. Patients ≥250 PNN/mm3 were compared with those with PNN <250 /mm3.
--> How could the authors ensure themselves without any analytical control?
Author Response
The manuscript has been well-improved regarding to my preivous comments. Nevertheless, some remains to be adressed.
Methods: how were the ascites collected (collection medium? delivery time?) The authors should describe the homogeneity/heterogeneity of the biological techniques used.
We added this paragraph in “Ascites sample analysis”: “For each patient in the PHRC and Paris cohorts, ascites samples were collected on the day of hospital admission. The cell count was determined automatically. The two cohorts’ samples were collected between 2010 and 2014. After centrifugation at à 5000g/min, the supernatants were frozen, stored at -80°C at each participating site, and subsequently (in 2015) analyzed centrally using MIR-FEWS.” (Lines 107 to 111).
--> Give details about the automates, the sample media.
Thank you for your comment. While revising our manuscript we noticed an mistake. We apologize for this.
The cell count was not done by an automated system but by a technician. The ascites samples were collected in dry sterile tubes.
We have made these changes in the article (lane 108 to 110).
Method: What were the positive/negative controls?
There was no positive/negative control. The endpoint was the accepted threshold of PNN ≥ 250/mm3. Patients ≥250 PNN/mm3 were compared with those with PNN <250 /mm3.
--> How could the authors ensure themselves without any analytical control?
Spectroscopic measurement does not involve any reagent. Positive or negative check of a reaction are therefore not relevant.
Infrared absorption spectra are recorded by relative measurement of the base, or background, signal of a sensor, and measurement of signal with a sample.
Hence, measure quality control is performed by :
1 – checking minimal amplitude, to ensure a proper signal, and absence of contamination in the background signal.
2 – validating sample measurement with minimal signal to noise ratio as well as relative signal intensity of elements absorption peaks compared to water peaks, water being present in all ascites liquids.
We have made these changes in the article (lane 121 to 126).
We hope you will enjoy this new version of our manuscript.
We thank you again for your guidance in improving our article.
Yours faithfully.
Marwin FARRUGIA
Round 3
Reviewer 1 Report
This manuscript has been completed following my previous remarks.
Nevertheless, The typos described are not trivial errors (automatic vs. manual) and should call into question the veracity of the results.
Moreover, even in the absence of reagent, any technique must be controlled from an analytical point of view to ensure the quality of the results obtained (known positive or known negative sample?).
Finally, analysis by human operators (however experienced they may be) raises new questions: double reading? double entry of results?